# Association of Psychosocial and Health Factors with Long COVID Symptoms in Students in Medicine-Related Departments: A Cross-Sectional Survey [note 1]

**DOI:** 10.3390/healthcare13151855

**Published:** 2025-07-30

**Authors:** Yu-Hsin Liu, Yi-Hsien Su, Su-Man Chang, Mei-Yu Chang, Wei-Fen Ma

**Affiliations:** 1School of Nursing, National Cheng Kung University, Tainan 701401, Taiwan; 2623cindy@gmail.com; 2In-Service Master Program of Interdisciplinary Long-Term-Care, School of Nursing, China Medical University, Taichung 406040, Taiwan; eva8111082@gmail.com (Y.-H.S.); manman930781@gmail.com (S.-M.C.); 3Department of Nursing, Taichung Veterans General Hospital, Taichung 407219, Taiwan; 4School of Nursing, China Medical University, Taichung 406040, Taiwan; 5Department of Nursing, China Medical University Hospital, Taichung 404327, Taiwan

**Keywords:** COVID-19, long COVID, medical-related students, medical university, psychophysiological health, Traditional Chinese Medicine

## Abstract

**Background**: As COVID-19 transitions to an endemic phase, long COVID symptoms remain a significant public health issue affecting both physical and mental health. A notable proportion of college students report symptoms such as fatigue, cough, and brain fog persisting for weeks or months post-infection. **Objectives**: This study explored the prevalence and contributing factors of long COVID symptoms among both infected and uninfected students in medicine-related departments. **Methods**: A cross-sectional study was conducted using online self-reported questionnaires completed by 1523 undergraduate and graduate students in medicine-related departments at a medical university. Participants who had tested positive for COVID-19 within the past three months were excluded. The survey assessed long COVID symptoms, with comparisons conducted between infected and uninfected groups. Multivariate logistic regression identified risk factors associated with long COVID symptoms. **Results**: Of the 1118 participants, 47.5% of those with a prior COVID-19 diagnosis reported long COVID symptoms within the past month. Significant differences between the infected and uninfected groups were observed in physical, cognitive, and psychological health. Logistic regression identified that prior COVID-19 diagnosis had an association with the presence of long COVID symptoms (odds ratio = 1.48, *p* = 0.024) after adjusted model analysis. Meanwhile, higher anxiety levels (odds ratio = 1.09, *p* < 0.001) and a BMI ≥ 24 (odds ratio = 4.50, *p* < 0.01) were identified as significant risk factors for post-infection syndrome among previously infected students. Sex and exercise habits also influenced symptom prevalence. **Conclusions**: Since late 2023, with those experiencing cumulative infections surpassing half of Taiwan’s population, long COVID symptoms have persisted as a widespread concern affecting both physical and mental health, continuing into 2025. This study underscores critical risk factors and symptom patterns among students in medicine-related departments, reinforcing the urgency of sustained surveillance and targeted interventions to facilitate comprehensive recovery.

## 1. Introduction

Since 15 April 2022, Taiwan has experienced a sharp increase in daily confirmed COVID-19 cases, with more than a thousand new cases reported daily. This surge has contributed to widespread public anxiety [1]. By 18 January 2023, the cumulative death toll in Taiwan had reached 15,929, while the global death toll was recorded at 6.78 million [1]. By the end of January 2023, the number of confirmed COVID-19 cases in Taiwan had reached 9,569,418, accounting for 40.9% of the total population (estimated at 23.4 million). Starting from 20 March 2023, the isolation requirement for mild COVID-19 cases was lifted, signaling a gradual transition toward managing COVID-19 as endemic, similarly to influenza. From 21 January 2020 to 20 March 2023, Taiwan reported a cumulative total of 10,236,886 confirmed cases, with a confirmation rate of 44.15% [2]. Although COVID-19 is now treated as an endemic virus, similarly to influenza, the pandemic and its long-term physical and psychological impacts remain significant. In particular, the persistent global burden of long COVID continues to warrant close attention.

Persistent symptoms such as brain fog and fatigue have been reported by many individuals even after recovering from COVID-19 [3]. The World Health Organization (WHO) classifies symptoms persisting for at least three months after an initial infection as long COVID [4,5]. Among students, reports of long COVID symptoms, including memory issues, fatigue, and breathing difficulties, have become increasingly common [6]. As the pandemic transitions into an endemic phase, these symptoms continue to pose significant challenges, impacting both physical and mental health [7]. While not immediately life-threatening, they can gradually lead to a deterioration in overall health, potentially leading to more serious conditions over time [8].

Managing post-infection symptoms to reduce the risk of severe complications remains a crucial healthcare priority [9]. However, as COVID-19 becomes endemic, long COVID symptoms have increasingly disrupted daily routines, often in subtle ways [5]. Changes in everyday life—such as school closures, online learning, restricted outings, and mandatory mask-wearing—have significantly altered individuals’ routines [10]. In Taiwan, in-person classes were gradually suspended beginning in April 2022 [2]. Research has also shown that students frequently report lingering symptoms such as cough, fatigue, and brain fog for weeks or months after recovering from the virus [10]. Therefore, the long-term effects of COVID-19 and the factors associated with these symptoms among student populations have become an urgent and pressing issue.

The pandemic has heightened people’s awareness of self-care, leading many individuals diagnosed with COVID-19 to explore alternatives to conventional medicine, including traditional Chinese medicines (TCMs) [11]. Among these, NRICM101, also known as Qing Guan Yi Hao [11], has gained widespread popularity [12]. NRICM101 is a multi-herb TCM formula. Prepared as a concentrated granule under Good Manufacturing Practice standards, it is intended to clear lung heat, detoxify the body, resolve phlegm, and prevent disease progression through synergistic effects [11,12]. NRICM101 has been integrated into comprehensive treatment approaches for COVID-19. Compared to conventional antiviral drugs, it is more readily accessible and was widely used during the peak of the pandemic [11,12,13]. Its potential role in alleviating long COVID symptoms has attracted increasing research interest.

Interestingly, individuals without prior COVID-19 infection also exhibit similar symptoms, such as fatigue, mood disturbances, and insomnia [14,15]. This observation raises the question of whether long COVID symptoms are solely attributable to SARS-CoV-2 infection. This evolving understanding highlights the need for further investigation into the nature and broader impact of long COVID symptoms [8]. To address this knowledge gap, in this study, we aimed to examine the relationship between long COVID symptoms by comparing two groups of medical university students from health-related disciplines: those with confirmed COVID-19 infections and those who had never been infected. The study objectives were as follows:To assess differences in self-reported physiological and psychological health between students with confirmed COVID-19 infections and uninfected individuals;To examine the prevalence and characteristics of long COVID symptoms among students after recovery;To identify factors associated with the occurrence of long COVID symptoms following infection;To identify factors influencing the occurrence of long COVID symptoms following COVID-19 infection.

The study framework is illustrated in the figure below (Figure 1).

## 2. Materials and Methods

### 2.1. Study Design and Participants

For this cross-sectional study, we recruited students from medicine-related departments from a medical university in central Taiwan through convenience sampling. Data collection was conducted between November 2022 and January 2023. Inclusion Criteria: Participants were officially registered undergraduate or graduate students with internet access who were able to complete the questionnaire and who provided written informed consent. Exclusion Criteria: Students were excluded if they had major physical or mental illnesses affecting their physiological or psychological health, gave incomplete responses, or had experienced significant physical health issues or a recent onset of psychiatric illness within the past six months.

Of the 1523 questionnaires distributed, 1118 met the inclusion criteria and were included in the final analysis. As COVID-19 was classified as a Category 5 notifiable infectious disease requiring mandatory reporting (until its reclassification to Category 4 on 1 May 2023, when reporting was no longer required), data on students’ confirmed COVID-19 diagnoses and infection dates were obtained and verified through records from the university’s health center.

### 2.2. Instruments and Measures

#### 2.2.1. Demographic Inventory

A self-designed inventory was created to collect demographic and health-related data, including age, sex, height, weight, COVID-19 history (e.g., number of vaccine doses received), diagnosis status and date, and the use of TCM NRICM101 (Qing Guan Yi Hao [13]). Information on severe physical disabilities (e.g., fractures) and mental illnesses such as schizophrenia, affective disorders, and major depression within the past six months was also gathered. The inventory ensured the comprehensive collection of data related to the study’s objectives. Apart from the history of students’ confirmed COVID-19 infections, which was uniformly verified through the university health center’s database once approval was obtained after the completion of data collection, all other information was self-reported by the participants. The English version has been uploaded as part of the Appendix A.

#### 2.2.2. Chinese Mandarin State and Trait Anxiety Inventory Form Y1 (CMSTAI-Y1)

The CMSTAI-Y1, translated by Ma et al. in 2008 [16], is a validated self-report scale assessing the participant’s level of anxiety through 20 items. Responses are rated on a 4-point Likert scale, with total scores ranging from 20 to 80. Higher scores indicate greater situational anxiety levels [16]. The scale demonstrates strong internal consistency (Cronbach’s α = 0.91–0.92) and test–retest reliability (r = 0.76–0.91) over two weeks [17]. Its frequent use in diverse Taiwanese populations, including university students [18,19,20], makes it suitable for this study. In this study, the internal consistency reliability of the CMSTAI-Y1 measured using Cronbach’s alpha was 0.95.

#### 2.2.3. Chinese Health Status Checklist for the Past Month

A self-designed checklist was developed to evaluate long COVID symptoms across six domains, each addressing specific health aspects [3,4,5,6]. The domains included physiological symptoms (19 items), cognitive performance (4 items), mental status (10 items), sleep condition (5 items), activity engagement (5 items), and changes in interpersonal relationships (3 items). The first three domains—physiological, cognitive, and mental status—comprised a total of 33 items. These items underwent content validation by a panel of five experts, including two infectious disease physicians, one psychiatrist, and two family medicine physicians. The experts evaluated whether the items aligned with the World Health Organization’s definitions of long COVID symptoms. The content validity index (CVI) for the checklist was 0.90, indicating high agreement among the experts. Participants were instructed to mark “V” next to the items that applied to their personal experiences. The English version has been uploaded as part of the Appendix A.

The physiological symptoms domain assessed a range of physical health issues, with options including no physical problems, cold/respiratory symptoms, fever, hypertension, headache/dizziness/head pressure, skin redness/itching/atopic dermatitis/eczema/urticaria, difficulty breathing, chest pain/tightness, heart palpitations, musculoskeletal or joint pain, hair loss, oral ulcers, the impairment of smell or taste, frequent diarrhea, decreased physical energy, weight gain, rapid weight loss, changes in menstrual cycle, and fatigue. In this study, the internal consistency reliability of the physiological domain, as measured using Cronbach’s alpha, was 0.68.

The cognitive performance items measured challenges such as difficulty concentrating, short-term memory decline, spatial orientation problems, and reduced comprehension ability. In the present study, the internal consistency reliability of this domain, as measured using Cronbach’s alpha, was 0.65. The mental status domain captured symptoms that could significantly impact daily life, with options including no mental issues, anxiety, perceived high stress, mood instability, unexplained low mood, a lack of motivation, depression, feelings of worthlessness or guilt, suicidal thoughts, and persistent negative self-perceptions. The Cronbach’s alpha for this domain was 0.83, indicating good internal consistency.

Activity engagement focused on exercise and physical activity patterns, with options including no exercise routine, regular ≥30 min sweating exercise 3–5 times per week, regular ≥30 min sweating exercise 1–2 times per week, occasional light activity (less than 30 min per session), and significantly reduced outdoor activities compared to pre-pandemic levels. Sleep-related issues were assessed, with options including no problems sleeping, difficulty falling asleep (taking at least 30 min to fall asleep), difficulty maintaining sleep (waking up 2 or more times during the night), feeling fatigued despite adequate sleep, and taking medication for sleep.

Finally, the questionnaire asked about changes in interpersonal relationships that were important to participants in order to evaluate changes in their relationships, which were categorized as no change, better, or worse.

### 2.3. Study Procedure and Ethical Considerations

This study was conducted in accordance with the principles outlined in the Declaration of Helsinki and received approval from the Institutional Review Board of China Medical University Hospital in Taiwan (CMUH111-REC3-147). Participants were informed about the study purpose and procedures before completing the survey. Written informed consent was obtained, with voluntary participation and privacy protection emphasized. Participants could withdraw at any time without consequences. Data confidentiality was strictly maintained, and no information was shared with third parties.

### 2.4. Data Analysis

Data were analyzed using SPSS version 29.0, with statistical significance set at alpha = 0.05. Descriptive statistics summarized the study variables. Normally distributed variables were assessed by skewness and kurtosis analysis. Group comparisons were conducted using independent *t*-tests for continuous variables and chi-squared or Fisher’s exact tests for categorical variables. The presence of long COVID symptoms was confirmed to be binary, and predictor variables included a combination of categorical and continuous variables; logistic regression analysis identified factors associated with long COVID symptoms. Model fit was assessed using Nagelkerke’s R^2^ and the criteria, proposed by Hosmer & Lemeshow and MedCalc [21], that R^2^ values between 0.1 and 0.2 indicate modest explanatory power and values between 0.2 and 0.4 indicate moderate explanatory power.

## 3. Results

### 3.1. Demographic Characteristics

A total of 1523 online surveys were completed, comprising 588 students with confirmed COVID-19 diagnoses and 935 without. After the exclusion of participants infected within the past three months (*n* = 405), 1118 (73.4%) individuals were included in the final analysis. The mean age of participants was 23.6 years (SD = 9.3), with females constituting the majority (67%, *n* = 749). The average body mass index (BMI) was 21.6 (SD = 3.8), reflecting a healthy weight range. The mean anxiety score was 44.1 (SD = 10.5), indicating low-to-moderate anxiety levels. Sleep disturbances were reported by 27.5% (*n* = 307) of participants, while 27.6% (*n* = 309) reported no regular exercise habits.

Participants were enrolled from various academic disciplines within the university. Specifically, 179 students (16.01%) were from the College of Medicine, 197 (17.62%) were from the College of Chinese Medicine, 199 (17.79%) were from the College of Pharmacy, 179 (16.01%) were from the College of Public Health, 192 (17.17%) were from the College of Health Care, 56 (5.01%) were from the College of Life Sciences, 21 (1.87%) were from the College of Humanities and Technology, 52 (4.65%) were from the College of Dentistry, and 43 (3.85%) were from the College of Biomedical Engineering.

Among the participants, 183 individuals (16.4%) had confirmed COVID-19 diagnoses from more than three months prior, while the remaining participants were non-diagnosed. Of the 183 diagnosed students, over half (53.3%, *n* = 95) reported using Qingguan No. 1 during their COVID-19 recovery. No statistically significant differences were found between diagnosed and non-diagnosed individuals in demographic characteristics, anxiety levels, sleep patterns, exercise behaviors, or interpersonal changes, suggesting homogeneity across most variables. However, a significant difference was observed in the frequency of exercising 1–2 times per week (α^2^ = 4.32, *p* = 0.035). Detailed demographic characteristics are presented in Table 1.

### 3.2. Physiological and Psychological Health Status

Among the 1118 survey participants, 47.5% of individuals with a COVID-19 diagnosis reported experiencing long COVID physical health symptoms within the past month, compared to 37.3% of non-diagnosed individuals. Significant differences in physiological health conditions were observed between the two groups, including respiratory symptoms, fever, breathing difficulties, chest pain or tightness, muscle and joint pain, and reduced physical strength (*p* < 0.05) (Table 2).

In terms of cognitive and psychological health, 40% of diagnosed individuals reported no recent issues, compared to 37.3% of non-diagnosed participants. However, significant differences were identified in short-term memory decline (*p* < 0.001) and mood instability, which approached statistical significance between the two groups (Table 3).

### 3.3. Impact of Long COVID Symptoms on Diagnosed Individuals

According to the WHO’s criteria for long COVID symptoms [4], the most common symptoms reported by students in this study were fatigue (27.7%) and sleep disturbances (19.4%). These findings indicate that long COVID symptoms are not limited to students diagnosed with COVID-19. A comparison of symptoms between diagnosed and non-diagnosed individuals revealed significant differences in fatigue, breathing difficulties, chest pain, and musculoskeletal pain, highlighting the role of COVID-19 infection in the manifestation of long COVID symptoms.

Logistic regression analysis revealed that for all students, being male, being older, not engaging in regular activities, and having higher anxiety levels were associated with an increased risk of developing long COVID symptoms. A confirmed COVID-19 diagnosis showed an association with the presence of long COVID symptoms (odds ratio = 1.48, *p* = 0.024) after adjusted model analysis. In addition, particular attention was given to the subgroup of previously infected students. Among this group, 148 students had a BMI < 24, with values ranging from 14.56 to 23.94 (mean = 20.03, SD = 2.03; skewness = –0.114, kurtosis = –0.592), and 35 students had a BMI ≥ 24, with values ranging from 24.09 to 37.24 (mean = 27.55, SD = 3.47; skewness = 1.164, kurtosis = 0.801). Among previously infected students, a BMI ≥ 24 (odds ratio = 4.50, *p* < 0.01) and higher anxiety levels (odds ratio = 1.09, *p* < 0.001) emerged as the most significant factors associated with an elevated risk of long COVID symptoms. Model fit was assessed using Nagelkerke’s R^2^ [21]. The adjusted model yielded a Nagelkerke R^2^ of 0.1597 for the total sample (*n* = 1118), indicating modest explanatory power. Subgroup analyses showed Nagelkerke R^2^ values of 0.1537 for the non-infected group (*n* = 935) and 0.2597 for the confirmed COVID-19 group (*n* = 183), reflecting acceptable model fit, particularly among the infected subgroup. Detailed results are presented in Table 4.

## 4. Discussion

### 4.1. Main Findings

This is the first study in Taiwan to explore the prevalence and risk factors associated with long COVID symptoms among students in medicine-related departments at a medical university. The findings revealed that 47.5% of students with a history of COVID-19 infection more than three months prior reported experiencing long COVID symptoms, with significant differences in physical and cognitive health compared to their uninfected peers. Interestingly, a substantial proportion of uninfected students also reported similar symptoms, suggesting that psychosocial factors, such as anxiety, may contribute to the subjective experience of long COVID symptoms.

The comparison focused on physiological symptoms, cognitive performance, psychological symptoms, sleep conditions, activity behaviors, and interpersonal relationship changes. There was an association after adjusted model analysis between a confirmed COVID-19 diagnosis and long COVID symptoms. This suggests that biological infection with lifestyle factors together may be sufficient to explain the occurrence of long COVID-like symptoms. Future research is warranted, incorporating a broader range of variables, in order to explore the potential mediating mechanisms underlying these symptom presentations.

### 4.2. Physiological, Cognitive, and Psychological Symptoms

Among diagnosed individuals, significant physiological symptoms included respiratory issues such as chest tightness, breathing difficulties, fever, chest pain, musculoskeletal pain, and reduced stamina. These findings are consistent with prior research identifying respiratory distress as a hallmark of long COVID. For example, Kikkenborg et al. [15] reported similar symptoms—such as headache, loss of appetite, and memory impairment—in a nationwide study. Similarly, O’Mahoney et al. [22], in their systematic review and meta-analysis, highlighted persistent respiratory symptoms, including dyspnea and chronic cough, as common long-term effects. These results underscore the importance of targeted interventions in managing lingering respiratory symptoms and improving quality of life during post-COVID-19 recovery. However, current assessments of long COVID symptoms predominantly rely on self-reported data, lacking objective physiological validation, particularly for symptoms such as fatigue and dyspnea, which are highly susceptible to psychological influences. Future research should integrate physiological measurements (e.g., pulmonary function tests, heart rate variability, and sleep monitoring) alongside psychological assessments (e.g., anxiety and stress inventories) to examine the interplay between psychological and physiological responses. Such multimodal approaches are critical in clarifying whether long COVID symptoms are primarily driven by the sequelae of SARS-CoV-2 infection or by interacting biopsychosocial factors.

Cognitive and psychological symptoms, particularly short-term memory impairment and fatigue, were prominent among participants. Fatigue and disrupted sleep, identified as common long COVID symptoms by O’Mahoney et al. [22], were similarly reported in this study. Ceban et al. [23] found that one third of individuals experienced fatigue lasting 12 weeks or longer post-diagnosis. The overlap of these symptoms among both infected and uninfected individuals raises critical questions regarding the interaction between psychological distress and somatic symptoms, challenging the traditional distinction of long COVID symptoms as solely infection-related. Meanwhile, higher anxiety levels were strongly associated with the presence of long COVID symptoms.

Logistic regression analysis revealed that each unit increase in anxiety score significantly increased the likelihood of experiencing these symptoms, irrespective of infection status. This aligns with previous findings indicating that anxiety exacerbates somatic complaints and fatigue [24,25]. While limited research has directly addressed the susceptibility of individuals with elevated anxiety to long COVID, this correlation warrants further investigation.

Academic pressure in Chinese society, particularly during the university years, has been identified as a significant contributor to fatigue and poor sleep quality [10,26].

In this study, over 40% of participants reported experiencing elevated levels of fatigue, with an average sleep duration of 6.85 ± 0.95 h. Factors such as stress, caffeine intake, and irregular sleep patterns, as noted by Wang et al. [14], further compounded these issues.

Although both BMI subgroups (BMI ≥ 24 and BMI < 24) among infected participants exhibited a normal distribution, the BMI ≥ 24 subgroup consisted of only 35 individuals, representing a relatively small sample size. This limitation may have contributed to an inflated odds ratio for long COVID symptoms within this subgroup and should be interpreted with caution. Nonetheless, low physical activity levels and high BMI were identified as significant risk factors for long COVID symptoms—findings that align with global evidence linking an elevated BMI to increased post-viral fatigue and systemic inflammation [24,27]. Regular physical activity may help in alleviating these symptoms by enhancing cardiovascular function, muscular endurance, and psychological resilience. However, the social distancing measures implemented during the pandemic significantly curtailed the opportunities for physical activity [10,25], likely contributing to elevated levels of generalized and social anxiety [28,29].

### 4.3. Use of NRICM101

In this study, more than half of the students who had been infected with COVID-19 for over three months reported using NRICM101 as a complementary therapy during the acute phase of infection. This suggests that the use of TCM remains a culturally embedded health-seeking behavior in Taiwan. Previous research has shown that older adults, individuals residing in rural areas, and those with chronic health conditions are more likely to use TCM to manage COVID-19-related symptoms [12]. Although Chang’s study reported no significant differences in mortality or severe physiological outcomes between COVID-19 patients who did and did not use NRICM101 [11], Tsai’s findings indicated that NRICM101 use was associated with shorter hospitalization durations and fewer days of mechanical ventilation in patients with initially severe disease [13]. However, our study lacks information on hospitalization history and the severity of acute symptoms, which limits the ability to assess the therapeutic effectiveness of NRICM101 in this population. Given its reported benefits in relieving symptoms such as cough, phlegm, fever, and muscle soreness [11], further research is needed to clarify its role in the management of long COVID symptoms and to inform its optimal integration into post-COVID-19 care strategies.

Although numerous studies have examined long COVID, the persistence of symptoms for more than three months following acute infection has been shown to affect multiple organ systems, resulting in organ damage and long-term functional impairment. This condition poses a significant burden not only for individual health but also for healthcare systems and national economies [30]. Notably, neuropsychiatric symptoms, such as anxiety and mood disturbances, have been reported in as many as 25% of individuals with long COVID and should not be overlooked [31], and this study reported similar findings. A recent systematic review and meta-analysis synthesizing data from 50 studies involving millions of participants confirmed that individuals previously infected with SARS-CoV-2 are at significantly increased risk for at least five distinct physiological and cognitive symptoms compared to uninfected controls [32]. Given that current treatment strategies remain primarily symptom-relieving and supportive in nature, these findings underscore the urgent need for multidimensional, evidence-based approaches to mitigating the long-term health impacts of COVID-19 [32].

### 4.4. Limitations

This study has several limitations. First, the cross-sectional design restricts the ability to establish causal relationships between the identified factors and the development of long COVID symptoms, underscoring the need for longitudinal studies to better understand symptom progression. Additionally, the reliance on self-reported data introduces the potential for recall bias or inaccuracies in participants’ responses, which may affect the validity of the findings; incorporating objective clinical data into future research could enhance the robustness of results. Moreover, as university students are younger and have frequent group interactions during classes, it is possible that some participants had undetected COVID-19 infections. This misclassification of uninfected participants could introduce bias in the analysis of factors influencing long COVID symptoms.

Additionally, this study recruited participants from a medical university, including students from all departments. Students in medicine-related programs may differ significantly from those in non-medical fields due to their unique academic schedules, intensive coursework, and greater exposure to clinical environments during the COVID-19 pandemic. These factors may influence both their risk of infection and their physical and mental health outcomes, thereby limiting the generalizability of the findings to non-medical student populations or the broader young adult population. Furthermore, the lack of information on participants’ vaccination status and timing represents an additional limitation that may have influenced symptom presentation and recovery trajectories, thereby affecting the study’s conclusions. Finally, given that the use of TCMs such as NRICM101 is a common health-seeking behavior among the Taiwanese population, we recommend that future research further explore the potential benefits of such interventions, particularly in the management of severe COVID-19 cases and their long-term outcomes.

## 5. Conclusions

This study sheds light on the prevalence and risk factors of long COVID symptoms among university students in medicine-related departments, revealing that 47.5% of those with a history of COVID-19 infection reported long-term symptoms, including respiratory and musculoskeletal issues, fatigue, and cognitive impairments. Importantly, higher anxiety levels, COVID-19 infection, a BMI ≥ 24, and a lack of physical activity emerged as significant predictors of long COVID symptoms, highlighting the interplay between physical and psychological health. The findings also suggest that long COVID symptoms are not exclusive to those diagnosed with COVID-19, with psychosocial factors potentially contributing to symptomatology. Given its established efficacy in alleviating acute symptoms, further research is necessary to explore its impact on long COVID and to develop tailored interventions to address the multifaceted health challenges faced by the students affected. This study underscores the importance of proactive health management, including mental health support and lifestyle modifications, to mitigate the long-term effects of COVID-19.

## Figures and Tables

**Figure 1 healthcare-13-01855-f001:**
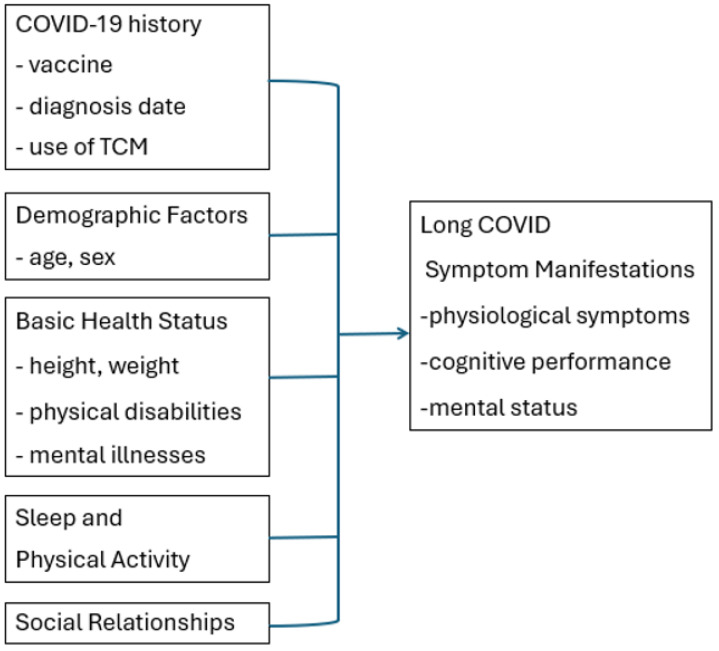
Study framework.

**Table 1 healthcare-13-01855-t001:** Comparing differences in *n* (%) in personal data, sleep, activity, and interpersonal changes.

Variables	All *n* = 1118	Not Confirmed with COVID-19 *n* = 935	Confirmed COVID-19 *n* = 183	*x*^2^/*t*	*p*
Age (M ± SD)	23.6 ± 9.3	23.8 ± 9.4	22.9 ± 8.7	1.09	0.275
Sex					
Female	749 (67.0)	631 (67.5)	118 (64.5)	0.85	0.653 ^†^
Male	368 (32.9)	303 (32.4)	65 (35.5)		
BMI (M ± SD)	21.6 ± 3.8	21.6 ± 3.8	21.5 ± 3.8	0.46	0.649
BMI < 18.5	218 (19.5)	183 (19.6)	35 (19.1)	0.44	0.802 ^†^
18.5 ≤ BMI < 24	668 (59.7)	555 (59.4)	113 (61.7)		
BMI ≥ 24	232 (20.8)	197 (21.1)	35 (19.1)		
Sleep patterns in the past month (multiple choice)					
No problems sleeping	811 (72.5)	688 (73.6)	123 (67.2)	3.12	0.077 ^†^
Difficulty falling asleep	129 (11.5)	108 (11.6)	21 (11.5)	0.001	0.977 ^†^
Difficulty maintaining sleep	89 (8.0)	74 (7.9)	15 (8.2)	0.02	0.897 ^†^
Fatigued even with adequate sleep	136 (12.2)	107 (11.4)	29 (15.8)	2.78	0.096 ^†^
Taking medication to sleep	23 (2.1)	19 (2.0)	4 (2.2)	--	0.781 ^‡^
Activity status in the past month (multiple choice)					
No exercise routine	309 (27.6%)	269 (28.8%)	40 (21.9%)	3.66	0.056 ^†^
Regular ≥30 min of sweating exercise ≥ 3 times per week	196 (17.5%)	155 (16.6%)	41 (22.4%)	3.59	0.058 ^†^
Regular ≥30 min of sweating exercise 1–2 times per week	320 (28.6%)	256 (27.4%)	64 (35.0%)	4.32	0.038 ^†^
≤30 min occasional walking or light activity	251 (22.5%)	215 (23.0%)	36 (19.7%)	0.97	0.325 ^†^
Significantly reduced outdoor activities	148 (13.2%)	127 (13.6%)	21 (11.5%)	0.59	0.442 ^†^
Interpersonal changes in the past month					
No change	963 (86.1%)	798 (85.3%)	165 (90.2%)	3.56	0.169 ^†^
Better	116 (10.4%)	101 (10.8%)	15 (8.2%)		
Worse	39 (3.5%)	36 (3.9%)	3 (1.6%)		

^†^ chi-squared test; ^‡^ Fisher’s exact test; BMI, body mass index.

**Table 2 healthcare-13-01855-t002:** Comparing differences in *n* (%) in physiological health in medical university students.

Variables	All *n* = 1118	Not Confirmed with COVID-19 *n* = 935	Confirmed COVID-19 *n* = 183	*x*^2^/*t*	*p*
No physical problems	682 (61.0)	586 (62.7)	96 (52.5)	6.71	0.010 ^†^
Cold/respiratory symptoms	166 (14.8)	123 (13.2)	43 (23.5)	12.95	<0.001 ^†^
Fever	38 (3.4)	25 (2.7)	13 (7.1)	9.15	0.002 ^†^
Hypertension	9 (0.8)	8 (0.9)	1 (0.5)	--	1.000 ^‡^
Headache/dizziness/head pressure	161 (14.4)	132 (14.1)	29 (15.8)	0.37	0.542 ^†^
Skin redness/itching/atopic dermatitis/eczema/urticaria	85 (7.6)	67 (7.2)	18 (9.8)	1.55	0.213 ^†^
Difficulty breathing	17 (1.5)	10 (1.1)	7 (3.8)	--	0.013 ^‡^
Chest pain/tightness	41 (3.7)	26 (2.8)	15 (8.2)	12.71	<0.001 ^†^
Heart palpitations	38 (3.4)	30 (3.2)	8 (4.4)	0.63	0.427 ^†^
Musculoskeletal or joint pain	41 (3.7)	28 (3.0)	13 (7.1)	7.32	0.007 ^†^
Hair loss	41 (3.7)	31 (3.3)	10 (5.5)	2.00	0.157 ^†^
Oral ulcers	22 (2.0)	20 (2.1)	2 (1.1)	--	0.560 ^‡^
Impairment of smell or taste	9 (0.8)	7 (0.7)	2 (1.1)	--	0.647 ^‡^
Frequent diarrhea	40 (3.6)	34 (3.6)	6 (3.3)	0.06	0.812 ^†^
Decreased physical energy	94 (8.4)	67 (7.2)	27 (14.8)	11.44	<0.001 ^†^
Weight gain	53 (4.7)	48 (5.1)	5 (2.7)	1.95	0.162 ^†^
Rapid weight loss	4 (0.4)	3 (0.3)	1 (0.5)	--	0.511 ^‡^
Changes in menstrual cycle	84 (7.5)	72 (7.7)	12 (6.6)	0.29	0.592 ^†^
Fatigue	283 (25.3)	222 (23.7)	61 (33.3)	7.45	0.006 ^†^

^†^ chi-squared test; ^‡^ Fisher’s exact test.

**Table 3 healthcare-13-01855-t003:** Comparing differences in *n* (%) in cognitive and psychological symptoms in medical university students.

Variables	All *n* = 1118	Not Confirmed with COVID-19 *n* = 935	Confirmed COVID-19 *n* = 183	*x*^2^/*t*	*p*
Cognitive Performance					
Difficulty concentrating	120 (10.7%)	94 (10.1%)	26 (14.2%)	2.76	0.097 ^†^
Short-term memory decline	119 (10.6%)	86 (9.2%)	33 (18.0%)	12.56	<0.001 ^†^
Challenges with spatial orientation	6 (0.5%)	6 (0.6%)	0 (0.0%)	--	0.597 ^‡^
Reduced comprehension ability	42 (3.8%)	32 (3.4%)	10 (5.5%)	1.77	0.184 ^†^
Psychological Symptoms					
No mental health issues	696 (62.3%)	586 (62.7%)	110 (60.1%)	0.43	0.513 ^†^
State Anxiety Scale (M ± SD)	44.1 ± 10.5	44.1 ± 10.6	43.9 ± 9.9	0.28	0.781
Anxiety	143 (12.8%)	125 (13.4%)	18 (9.8%)	1.71	0.191 ^†^
Perceived stress	165 (14.8%)	142 (15.2%)	23 (12.6%)	0.83	0.361 ^†^
Mood instability	87 (7.8%)	79 (8.4%)	8 (4.4%)	3.55	0.060 ^†^
Unexplained low mood	76 (6.8%)	62 (6.6%)	14 (7.7%)	0.25	0.616 ^†^
Lack of motivation	122 (10.9%)	99 (10.6%)	23 (12.6%)	0.62	0.432 ^†^
Depression	52 (4.7%)	45 (4.8%)	7 (3.8%)	0.34	0.562 ^†^
Feelings of worthlessness or guilt	32 (2.9%)	26 (2.8%)	6 (3.3%)	0.14	0.712 ^†^
Suicidal thoughts	15 (1.3%)	12 (1.3%)	3 (1.6%)	--	0.723 ^‡^
Persistent negative self-perception	51 (4.6%)	44 (4.7%)	7 (3.8%)	0.27	0.602 ^†^

^†^ chi-squared test; ^‡^ Fisher’s exact test.

**Table 4 healthcare-13-01855-t004:** Logistic regression analysis of factors influencing long COVID syndrome.

Variables	Unadjusted Model (*n* = 1118)		Adjusted Model (*n* = 1118)		Not Confirmed with COVID-19 (*n* = 935)		Confirmed COVID-19 (*n* = 183)	
	OR [95%CI]	*p*	OR [95%CI]	*p*	OR [95%CI]	*p*	OR [95%CI]	*p*
Sex								
Female	Ref.		Ref.		Ref.		Ref.	
Male	0.76 [0.59–0.98]	0.034	0.87 [0.66–1.16]	0.355	0.85 [0.62–1.16]	0.312	0.98 [0.46–2.08]	0.956
Age (years)	1.01 [1.00–1.03]	0.036	1.01 [1.00–1.03]	0.055	1.01 [1.00–1.03]	0.115	1.02 [0.98–1.07]	0.325
BMI								
<24	Ref.		Ref.		Ref.		Ref.	
≥24	1.21 [0.91–1.62]	0.189	1.27 [0.91–1.75]	0.156	1.04 [0.73–1.48]	0.821	4.50 [1.73–11.71]	0.002
Vaccine Doses								
0	Ref.		Ref.		Ref.		Ref.	
1	0.78 [0.19–3.19]	0.727	0.88 [0.20–3.89]	0.870	0.54 [0.11–2.79]	0.464	--	0.990
2	0.50 [0.19–1.32]	0.161	0.75 [0.27–2.07]	0.585	0.59 [0.18–1.96]	0.388	1.21 [0.14–10.28]	0.864
≥3	0.49 [0.20–1.20]	0.119	0.62 [0.25–1.59]	0.324	0.46 [0.15–1.43]	0.182	1.43 [0.23–8.69]	0.700
COVID-19 Diagnosis								
No	Ref.		Ref.					
Yes	1.36 [0.99–1.87]	0.055	1.48 [1.05–2.09]	0.024				
Number of Times Per Week Undertaking Regular Activity in the Past Month (multiple choice)								
≥3	Ref.		Ref.		Ref.		Ref.	
1–2	1.32 [0.91–1.91]	0.143	1.25 [0.84–1.86]	0.269	1.11 [0.71–1.73]	0.660	2.16 [0.86–5.41]	0.102
No	1.55 [1.11–2.16]	0.010	1.30 [0.90–1.87]	0.165	1.17 [0.78–1.77]	0.443	2.23 [0.90–5.51]	0.083
State Anxiety Scale	1.07 [1.06–1.09]	<0.001	1.07 [1.06–1.09]	<0.001	1.07 [1.06–1.09]	<0.001	1.09 [1.04–1.13]	<0.001

OR, odds ratio; CI, confidence interval.

## Data Availability

These study data are deidentified participant data. The data that support the findings of this study will be available beginning 12 months and ending 36 months following the article publication upon reasonable request to the corresponding author, WFM, at lhdaisy@mail.cmu.edu.tw.

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
