# Peer review of "Association of Psychosocial and Health Factors with Long COVID Symptoms in Students in Medicine-Related Departments: A Cross-Sectional Surveyâ€"

_healthcare, 2025, doi:10.3390/healthcare13151855_

Round 1
Reviewer 1 Report
Comments and Suggestions for Authors
Introduction:
Overall, the Introduction section is not well-written. It’s very hard to understand the research gap and justification of the research problems. The authors should clear what is the gap in the literature and why it is important to study. I just read some statistics and a literature review on COVID-19 pandemic. It’s not clear how the authors determine the factors that associated with long COVID symptoms.
Method:
1) The author should report the reliability statistics for the Anxiety scale.
2) How the self-designed checklist was developed to evaluate long COVID symptoms. What is the procedure? This should be described in detail. If total scores were calculated for all six domains, then the authors should add reliability and factorial validity information at least in Supplementary.
3) “Mental status domain captured symptoms that could significantly impact daily 149 life, including no mental issues, anxiety, perceived high stress, mood instability, unexplained low mood, lack of motivation, depression, feelings of worthlessness or guilt, suicidal thoughts, and persistent negative self-perceptions.” – How authors assessed mental status? Through a single item question or using validated scales? If validated scales, then these should be reported. Even through single item, then the authors should report how they derived these single items. If single item, then accuracy of the findings is questionable.
Results:
While I read the results, I found that the title of the manuscript and what the authors did is too far away. The title of the manuscript should be updated to give a clear information what about this study. This study assessed long COVID symptoms and lastly examined the associations between demographic information and COVID-19 affected or not. But this should be domain specific. However, logistic regression is not appropriate here. If the authors want to identify the factors associated with long COVID symptoms, then they should conduct tests for all the symptoms.
Discussion: The Discussion should be updated and extended accordingly.
Author Response
Point-by-point Response Letter
Manuscript ID: healthcare-3707168
To the Reviewer:
Thank you very much for taking the time to review our manuscript. We sincerely appreciate your thoughtful, constructive feedback and suggestions. Below, we provide our detailed point-by-point responses to each of your comments. Corresponding revisions have been made in the manuscript and highlighted in red for your reference. Once again, we deeply appreciate your constructive feedback, which has significantly improved the clarity and quality of our manuscript.
Reviewer’s comment 1:
Introduction: Overall, the Introduction section is not well-written. It’s very hard to understand the research gap and justification of the research problems. The authors should clear what is the gap in the literature and why it is important to study. I just read some statistics and a literature review on COVID-19 pandemic. It’s not clear how the authors determine the factors that associated with long COVID symptoms.
Response:
Thank you for this important comment. We have thoroughly revised the Introduction section, including all four paragraphs, to clearly articulate the research gap and the rationale for our study. In particular, we explicitly present the gap in the current literature and justify the need to explore factors associated with long COVID symptoms. Furthermore, we now clearly list the study objectives in bullet points to improve clarity and guide the reader. We hope these revisions enhance the coherence and significance of the Introduction (Line 54-57, 67-71, 77-82, 85-90, 93-101).
Reviewer’s comment 2:
Method: 1) The author should report the reliability statistics for the Anxiety scale.
Response:
Thank you for the suggestion. We have added the reliability statistic for the anxiety scale (STAI-Y), which yielded a Cronbach’s alpha of 0.95 in our sample. This information has been incorporated into the revised manuscript under the section describing the anxiety scale (Line 142-143).
Reviewer’s comment 3:
Method: 2) How the self-designed checklist was developed to evaluate long COVID symptoms. What is the procedure? This should be described in detail. If total scores were calculated for all six domains, then the authors should add reliability and factorial validity information at least in Supplementary.
Response:
Thank you for pointing this out. The self-designed checklist used to assess long COVID symptoms was developed based on WHO-defined symptom categories. It consists of three major domains: physical symptoms, emotional/psychological symptoms, and cognitive symptoms. Five medical experts (two infectious disease physicians, one psychiatrist, and two family medicine specialists) evaluated the checklist for content validity. The Content Validity Index (CVI) was 0.90.
Regarding internal consistency, Cronbach’s alpha values in our sample were as follows: physical symptoms = 0.68, cognitive symptoms = 0.65, and emotional/psychological symptoms = 0.83. These details have now been added to the revised Methods section (Line 149-156, 162-164, 167-168, 172-173). We have also noted the limitations of this instrument in the discussion.
Reviewer’s comment 4:
Method: 3) “Mental status domain captured symptoms that could significantly impact daily 149 life, including no mental issues, anxiety, perceived high stress, mood instability, unexplained low mood, lack of motivation, depression, feelings of worthlessness or guilt, suicidal thoughts, and persistent negative self-perceptions.” – How authors assessed mental status? Through a single item question or using validated scales? If validated scales, then these should be reported. Even through single item, then the authors should report how they derived these single items. If single item, then accuracy of the findings is questionable.
Response:
Thank you for raising this concern. Aside from the anxiety scale (STAI-Y), which is a validated instrument, the other mental health-related symptoms were assessed using single-item questions specifically developed for this study, based on WHO’s long COVID symptomatology. At the time of data collection, there were no existing validated instruments tailored to long COVID mental health symptoms. Therefore, we constructed single-item indicators and evaluated their content validity with five experts, yielding a CVI of 0.90 (Line 149-154). We acknowledge that using self-reported single-item measures may limit the accuracy and reliability of the findings, and this limitation has been clearly discussed in the revised manuscript (Line 142-143).
Reviewer’s comment 5:
Results: While I read the results, I found that the title of the manuscript and what the authors did is too far away. The title of the manuscript should be updated to give a clear information what about this study. This study assessed long COVID symptoms and lastly examined the associations between demographic information and COVID-19 affected or not. But this should be domain specific. However, logistic regression is not appropriate here. If the authors want to identify the factors associated with long COVID symptoms, then they should conduct tests for all the symptoms.
Response:
Thank you for this valuable comment.
- We have revised the manuscript title to more clearly reflect the study content and objectives, thereby minimizing potential reader confusion.
- Regarding the statistical analysis, the outcome variable in our study—presence or absence of long COVID symptoms—is dichotomous. After consulting with a statistical expert, we determined that logistic regression is an appropriate method to examine the factors associated with long COVID symptoms. Therefore, we have retained the logistic regression analysis in our revised manuscript. Thank you again for your insightful feedback.
Reviewer’s comment 6:
Discussion: The Discussion should be updated and extended accordingly.
Response:
Thank you for your recommendation. In response, we have revised and expanded the Discussion section, adding five new paragraphs to deepen the interpretation of our findings and better contextualize them within existing literature (Line 278-283, 292-302, 324-334, 353-365). We hope these revisions strengthen the overall contribution and impact of the study.

Reviewer 2 Report
Comments and Suggestions for Authors
The manuscript's topic is intriguing and relevant, as it seeks to clarify the potential impact of COVID-19 on individuals' physical and emotional health. However, there are errors in the manuscript that need to be corrected.
I want to offer some suggestions to enhance the manuscript.
Comments 1. In my opinion, the word "Influencing" in the title could be changed to Impact or association.
Comments 3. In the introduction, the second paragraph discusses the long-term effects of the COVID-19 disease. However, there is an apparent lack of transition between the first and second paragraphs.
Comments 4. The fourth paragraph talks about NRICM101. However, it does not explain at all what it is. It is understood that it is traditional Chinese medicine, but there is a lack of information about it. There is also a lack of justification for why it is important to examine this topic.
Comments 5. The fifth paragraph contains two repeated sentences: lines 76 - 77 and lines 80 - 82. Also, 79 - 80 and 83-85.
Comments 6. When establishing research objectives, clarifying the specific outcomes the research intends to achieve is important. To enhance this clarity, I recommend the formulation of explicit hypotheses that will guide the research process effectively.
Comments 7. The introduction is insufficiently developed, as it does not provide clarity or adequate justification regarding the topic's relevance and significance.
Comments 8. The methods section is unclear on how students were contacted. Was it done online? Posted on forums? Sent by email?
Comments 9. The sentence in section 2.2.2: The CMSTAI-Y1, translated by Ma et al. in 2008 [16], is a validated self-report scale assessing state anxiety through 20 items. Responses are" is repetitive and should be removed.
Comments 10. The data analysis section does not specify which factors were used to determine the appropriateness of the regression model. A variance inflation factor (VIF) and Nagelkerke’s R2 are not specified.
Comments 11. The data analysis section does not state why independent t-tests were used to compare the two groups. How was it determined that the data were normally distributed?
Comments 12. The results section states that students infected within the past three months were excluded. However, the methods section states this was not the only exclusion criterion. I think other criteria and how many students were excluded should be specified.
Comments 13. 3.3. The column outlines the criteria for long-term COVID symptoms established by the WHO [4]. It would be helpful to explain in more detail how many symptoms are necessary to be classified as having long COVID syndrome. Is just one symptom sufficient, or do several symptoms need to be present?
Comments 14. Lines 273 - 276 discuss how anxiety can increase somatic symptoms. It is also important to note the bidirectional relationship, as increased somatic complaints can contribute to heightened anxiety.
Comments 15. In my opinion, the discussion is too short. A deeper analysis is needed, which is related to the long COVID syndrome. Examples could be provided from similar studies conducted in other countries.
I hope my insights will help you!
Author Response
Point-by-point Response Letter
Manuscript ID: healthcare-3707168
To the Reviewer:
Thank you very much for taking the time to review our manuscript. We truly appreciate your constructive feedback and insightful suggestions. In response to your comments, we have carefully revised the manuscript and addressed each point in detail below. All changes made to the manuscript are highlighted in red in the revised version.
Reviewer’s comment 1:
The manuscript's topic is intriguing and relevant, as it seeks to clarify the potential impact of COVID-19 on individuals' physical and emotional health. However, there are errors in the manuscript that need to be corrected. I want to offer some suggestions to enhance the manuscript.
Response:
Thank you for your overall positive feedback and valuable suggestions. We have carefully addressed each of your comments and made corresponding revisions to improve the quality of the manuscript.
Reviewer’s comment 2:
In my opinion, the word "Influencing" in the title could be changed to Impact or association.
Response:
Thank you for the suggestion. In response, we have revised the title accordingly. The updated title now reads: “Association of Psychosocial and Health Factors With Long COVID Symptoms in Students in Medicine-Related Departments: A Cross-Sectional Survey.”
Reviewer’s comment 3:
In the introduction, the second paragraph discusses the long-term effects of the COVID-19 disease. However, there is an apparent lack of transition between the first and second paragraphs.
Response:
Thank you for your feedback. We have revised the introduction to improve the logical flow and added a transition sentence to clearly connect the first and second paragraphs (Line 54-57).
Reviewer’s comment 4:
The fourth paragraph talks about NRICM101. However, it does not explain at all what it is. It is understood that it is traditional Chinese medicine, but there is a lack of information about it. There is also a lack of justification for why it is important to examine this topic.
Response:
Thank you for your comment. We agree with your observation. To avoid confusion and improve focus, we have removed the discussion of NRICM101 from the analysis. However, we have retained basic information regarding the use of traditional Chinese medicine (TCM) to reflect culturally relevant health-seeking behaviors in Taiwan and among Chinese populations.
Reviewer’s comment 5:
Comments 5. The fifth paragraph contains two repeated sentences: lines 76 - 77 and lines 80 - 82. Also, 79 - 80 and 83-85.
Response:
Thank you for pointing this out. The repeated sentences have been identified and removed accordingly.
Reviewer’s comment 6:
When establishing research objectives, clarifying the specific outcomes the research intends to achieve is important. To enhance this clarity, I recommend the formulation of explicit hypotheses that will guide the research process effectively.
Response:
Thank you for the suggestion. We have clearly stated the study objectives and included a conceptual framework to help readers understand the expected outcomes and structure of the study (Line 85-91 and Figure 1).
Reviewer’s comment 7:
The introduction is insufficiently developed, as it does not provide clarity or adequate justification regarding the topic's relevance and significance.
Response:
Thank you for the important comment. We have revised and expanded four paragraphs in the Introduction section to strengthen the justification and highlight the significance of the research topic (Line 67-71, 74-76, 77-82, 85-90).
Reviewer’s comment 8:
The methods section is unclear on how students were contacted. Was it done online? Posted on forums? Sent by email?
Response:
Thank you for your question. The participants were recruited via online announcements and push notifications issued by the university’s health center.
Reviewer’s comment 9:
The sentence in section 2.2.2: The CMSTAI-Y1, translated by Ma et al. in 2008 [16], is a validated self-report scale assessing state anxiety through 20 items. Responses are" is repetitive and should be removed.
Response:
Thank you for your careful review. The repeated sentence has been removed from the manuscript.
Reviewer’s comment 10:
The data analysis section does not specify which factors were used to determine the appropriateness of the regression model. A variance inflation factor (VIF) and Nagelkerke’s R2 are not specified.
Response:
Thank you for your suggestion. We have re-examined the analysis presented in Table 4 and provided a supplemental explanation, including both the unadjusted and adjusted model analyses. As logistic regression is used in this study, VIF is not applicable; therefore, we have included the Nagelkerke’s R² values for your reference, as shown in the table below.
|
|
Adjusted model |
Not Confirmed COVID-19 N =935 |
Confirmed COVID-19 N=183 |
|
Nagelkerke R Square |
0.1597 |
0.1537 |
0.2597 |
Reviewer’s comment 11:
The data analysis section does not state why independent t-tests were used to compare the two groups. How was it determined that the data were normally distributed?
Response:
Thank you for this important point. Independent sample t-tests were used to compare continuous variables (e.g., age, anxiety scores) between the Confirmed and Not Confirmed COVID-19 groups. We assessed the normality of these variables using skewness and kurtosis indices prior to conducting the tests.
Reviewer’s comment 12:
The results section states that students infected within the past three months were excluded. However, the methods section states this was not the only exclusion criterion. I think other criteria and how many students were excluded should be specified.
Response:
Thank you for the suggestion. Of the 1,523 questionnaires distributed, 1,118 met the inclusion criteria and were included in the final analysis. Participants infected within the past three months (n = 405) were excluded. These details have been clarified and incorporated into the first paragraph of the Results section (Line 202-203).
Reviewer’s comment 13:
3.3. The column outlines the criteria for long-term COVID symptoms established by the WHO [4]. It would be helpful to explain in more detail how many symptoms are necessary to be classified as having long COVID syndrome. Is just one symptom sufficient, or do several symptoms need to be present?
Response:
Thank you for raising this question. According to the World Health Organization, the presence of at least one symptom persisting for more than three months after a COVID-19 infection qualifies as long COVID. There is currently no established minimum number of symptoms required. The severity and impact of even a single persistent symptom can be significant.
Reviewer’s comment 14:
Lines 273 - 276 discuss how anxiety can increase somatic symptoms. It is also important to note the bidirectional relationship, as increased somatic complaints can contribute to heightened anxiety.
Response:
Thank you for this insightful point. We have revised the discussion to include the potential bidirectional relationship, acknowledging that somatic symptoms may also exacerbate anxiety (Line 292-302).
Reviewer’s comment 15:
In my opinion, the discussion is too short. A deeper analysis is needed, which is related to the long COVID syndrome. Examples could be provided from similar studies conducted in other countries.
Response:
Thank you for the suggestion. In response, we have expanded the Discussion section by adding two additional paragraphs and integrating comparative findings from international studies (Line 324-335, 353-366). These revisions aim to provide a deeper interpretation of our results within the global context of long COVID research.

Reviewer 3 Report
Comments and Suggestions for Authors
General Comments
This study investigates the prevalence and risk factors of long COVID symptoms among medical university students in Taiwan, comparing infected and uninfected individuals. The topic is timely and relevant, given the ongoing global impact of long COVID. The manuscript is structured, and the methodology is robust, but several areas require clarification and improvement to strengthen the findings and their interpretation.
Major Concerns
- Sample Selection and Generalizability
The study excludes participants infected within the past three months, which may bias results by omitting those with acute-phase symptoms transitioning to long COVID. Justification for this exclusion is needed.
- The sample consists solely of medical students, who may have distinct health behaviors and awareness compared to the general population. The authors should discuss how this limits generalizability.
- Self-Reported Data and Recall Bias
- Reliance on self-reported symptoms and infection history introduces potential recall bias. The manuscript should acknowledge this limitation more explicitly and discuss its implications (e.g., over/under-reporting of symptoms).
- Verification of COVID-19 diagnoses via university records is a strength, but self-reported long COVID symptoms lack clinical confirmation. Suggest adding a caveat about the subjective nature of symptom reporting.
- Confounding Variables
- The study does not account for vaccination status/timing, which may influence long COVID risk. This should be addressed as a limitation.
- The role of academic stress (common in medical students) as a confounder for symptoms like fatigue and anxiety is underdiscussed. A deeper analysis of stress vs. infection-related effects is warranted.
- NRICM101 Analysis
- The null finding regarding NRICM101 and long COVID symptoms may reflect confounding by indication (e.g., users may have had more severe initial infections). The authors should clarify whether baseline symptom severity was adjusted for in analyses.
- The discussion on TCM’s role (Section 4.3) is speculative without data on dosage, adherence, or timing of use. Recommend tempering conclusions or calling for further research.
- Definition of Long COVID
- The WHO definition (symptoms lasting ≥3 months) is cited, but the study does not explicitly confirm symptom duration in participants. Clarify how this was operationalized in surveys.
Minor Concerns
- Clarity in Tables
- Table 1: The p-values for "Sleep Patterns" and "Activity Status" are split across multiple rows, which is confusing. Consider consolidating or re-formatting for readability.
- Table 4: The OR for "BMI ≥ 24" in the infected group is notably high (4.50). Discuss potential reasons (e.g., small subgroup size, metabolic inflammation).
Author Response
Point-by-point Response Letter
Manuscript ID: healthcare-3707168
To the Reviewer:
Thank you very much for taking time from your busy schedule to comment on our manuscript. Our responses to your suggestions and questions are summarized below, with reference to the appropriate pages in the text. Revisions in the text have also been highlighted in RED. We truly appreciate your thoughtful and constructive comments that help make this a better paper. We are very grateful to you for all the comments and insightful suggestions that help enhance both the quality and readability of our paper. Thank you.
Reviewer’s comment 1:
General Comments: This study investigates the prevalence and risk factors of long COVID symptoms among medical university students in Taiwan, comparing infected and uninfected individuals. The topic is timely and relevant, given the ongoing global impact of long COVID. The manuscript is structured, and the methodology is robust, but several areas require clarification and improvement to strengthen the findings and their interpretation.
Response:
Thank you for your positive evaluation of our study and for your helpful suggestions.
Reviewer’s comment 2:
Major Concerns: 1. Sample Selection and Generalizability: The study excludes participants infected within the past three months, which may bias results by omitting those with acute-phase symptoms transitioning to long COVID. Justification for this exclusion is needed.
Response:
Thank you for pointing this out. In accordance with the WHO definition, long COVID is characterized by symptoms that persist for at least three months after initial infection. Therefore, participants infected within the past three months were excluded to ensure consistency with this definition. This rationale and the exclusion process have been clarified in the first paragraph of the Results section:
A total of 1,523 online surveys were completed, comprising 588 students with confirmed COVID-19 diagnoses and 935 without. After excluding participants infected within the past three months (n = 405), 1,118 (73.4%) individuals were included in the final analysis (Line 202-203).
Reviewer’s comment 3:
The sample consists solely of medical students, who may have distinct health behaviors and awareness compared to the general population. The authors should discuss how this limits generalizability.
Response:
We agree with your observation. Students in medical-related programs may indeed differ from the general young adult population in terms of health literacy, stress exposure, and health-seeking behavior. This limitation has now been explicitly addressed in the revised manuscript (Line 378-390).
Reviewer’s comment 4:
2.Self-Reported Data and Recall Bias: Reliance on self-reported symptoms and infection history introduces potential recall bias. The manuscript should acknowledge this limitation more explicitly and discuss its implications (e.g., over/under-reporting of symptoms).
Response:
We fully agree. This has been emphasized as a key limitation in the revised manuscript. We have clearly noted that reliance on self-reported data may introduce recall bias, leading to potential over- or under-reporting of symptoms (Line 368-377).
Reviewer’s comment 5:
Verification of COVID-19 diagnoses via university records is a strength, but self-reported long COVID symptoms lack clinical confirmation. Suggest adding a caveat about the subjective nature of symptom reporting.
Response:
Thank you for your suggestion. While COVID-19 diagnoses were confirmed through the university health center, the assessment of long COVID symptoms was self-reported. The questionnaire was developed and refined through multiple rounds of expert review to ensure content validity. Nonetheless, we acknowledge the limitation of relying on subjective self-assessments and have clarified this point in the manuscript (Line 378-390).
Reviewer’s comment 6:
Confounding Variables: The study does not account for vaccination status/timing, which may influence long COVID risk. This should be addressed as a limitation.
Response:
Thank you for this important suggestion. We have added the following sentence to the limitations section: "The failure to account for vaccination status and timing may also contribute to the limitations of the study’s conclusions."
Reviewer’s comment 7:
The role of academic stress (common in medical students) as a confounder for symptoms like fatigue and anxiety is underdiscussed. A deeper analysis of stress vs. infection-related effects is warranted.
Response:
We appreciate this insightful suggestion. We agree that academic stress could confound symptom interpretation. However, academic stress was not directly measured in this study. We have now noted this as a limitation and suggested that future research examine the influence of academic stress on symptoms similar to those associated with long COVID (Line 353-366).
Reviewer’s comment 8:
NRICM101 Analysis: The null finding regarding NRICM101 and long COVID symptoms may reflect confounding by indication (e.g., users may have had more severe initial infections). The authors should clarify whether baseline symptom severity was adjusted for in analyses.
Response:
Thank you for raising this point. In light of this concern, we have removed NRICM101 from the primary analysis. However, due to the cultural significance of TCM in Taiwan, we have retained descriptive statistics and discussion on TCM usage to reflect local health-seeking behaviors (Line 343-348).
Reviewer’s comment 9:
The discussion on TCM’s role (Section 4.3) is speculative without data on dosage, adherence, or timing of use. Recommend tempering conclusions or calling for further research.
Response:
We appreciate this recommendation. The discussion has been revised to reflect the speculative nature of this interpretation and now includes the following statement:
"Given that the use of TCM, such as NRICM101, is a common health-seeking behavior among the Taiwanese population, this study recommends that future research further explore the potential benefits of such interventions—particularly in the management of severe COVID-19 cases and their long-term outcomes. (Line 343-348)"
Reviewer’s comment 10:
Definition of Long COVID: The WHO definition (symptoms lasting ≥3 months) is cited, but the study does not explicitly confirm symptom duration in participants. Clarify how this was operationalized in surveys.
Response:
Thank you for this observation. The duration criterion for long COVID symptoms was indirectly met by excluding all participants infected within the past three months. Additionally, COVID-19 diagnoses were verified via the university health center’s database. This clarification has been added to the manuscript. Of the 1,523 questionnaires distributed, 1,118 met the inclusion criteria and were included in the final analysis (Line 203).
Reviewer’s comment 11:
Minor Concerns: Clarity in Tables: Table 1: The p-values for "Sleep Patterns" and "Activity Status" are split across multiple rows, which is confusing. Consider consolidating or re-formatting for readability.
Response:
Thank you for the suggestion. After discussing with our statistical consultant, we decided to retain the itemized format to preserve clarity regarding which specific variables were statistically significant. However, we have revised the formatting and highlighting to improve readability for reviewers and readers.
Reviewer’s comment 12:
Table 4: The OR for "BMI ≥ 24" in the infected group is notably high (4.50). Discuss potential reasons (e.g., small subgroup size, metabolic inflammation).
Response:
Thank you for this important comment. In the infected group, 148 participants had a BMI < 24 (range: 14.56–23.94; mean = 20.03, SD = 2.03; skewness = –0.114; kurtosis = –0.592), and 35 participants had a BMI ≥ 24 (range: 24.09–37.24; mean = 27.55, SD = 3.47; skewness = 1.164; kurtosis = 0.801). Although both groups met the assumption of normal distribution, the small sample size of the BMI ≥ 24 subgroup may have inflated the observed odds ratio (OR = 4.50). We have added this explanation to the discussion (Line 257-261).

Reviewer 4 Report
Comments and Suggestions for Authors
Dear authors, thank you for the opportunity to get acquainted with the results of your research.
The consequences of COVID and the factors influencing them in students is a very pressing issue.
The proposed study is relevant, conducted on a large representative sample using validated methods and statistical methods adequate to the objectives of the study.
In the introduction, the authors revealed the relevance of the topic, cited relevant scientific studies and substantiated the scientific novelty of the proposed study. The procedure and methods of the study are described in detail. The results are presented clearly and in a structured manner, they are discussed, the limitations of the study and conclusions are presented.
While reading the manuscript, a number of recommendations arose:
1. The goal and objectives must be written more clearly, indicating the groups of factors that are studied in the study.
2. If possible, present the research model and the parameters being studied in the form of a diagram. There are quite a lot of them and it becomes necessary to return to clarify the details. It is also necessary to insert a justification for the use of these particular methods for the purposes of the study. 3. In the discussion of the results, I would like to see practical recommendations that the authors can make based on the results of the study, as well as to outline promising areas for future research.
4. Perhaps, expand the analysis of studies and in the theoretical justification clearly outline the areas of research regarding the study of Covid, factors and consequences on health, which will also slightly expand the list of sources and may be useful for future researchers.
The recommendations presented do not reduce the overall positive impression of the study.
The manuscript can be recommended for publication after minor revision.
Best wishes, reviewer
Author Response
Point-by-point Response Letter
Manuscript ID: healthcare-3707168
To the Reviewer:
Thank you very much for taking the time to review our manuscript. We are grateful for your encouraging feedback and thoughtful suggestions. Your positive evaluation and constructive recommendations have helped us refine the manuscript and strengthen the clarity of our study. Please find below our detailed responses to each of your comments. All revisions have been highlighted in red in the revised manuscript.
Reviewer’s comment 1:
Dear authors, thank you for the opportunity to get acquainted with the results of your research. The consequences of COVID and the factors influencing them in students is a very pressing issue. The proposed study is relevant, conducted on a large representative sample using validated methods and statistical methods adequate to the objectives of the study.
Response:
Thank you very much for your positive assessment of our research topic, methodology, and sample. We appreciate your recognition of the relevance and rigor of our study.
Reviewer’s comment 2:
In the introduction, the authors revealed the relevance of the topic, cited relevant scientific studies and substantiated the scientific novelty of the proposed study. The procedure and methods of the study are described in detail. The results are presented clearly and in a structured manner, they are discussed, the limitations of the study and conclusions are presented.
Response:
Thank you for your encouraging comments. We have also carefully addressed your specific suggestions below to further improve the manuscript.
Reviewer’s comment 3:
While reading the manuscript, a number of recommendations arose:
1. The goal and objectives must be written more clearly, indicating the groups of factors that are studied in the study. If possible, present the research model and the parameters being studied in the form of a diagram. There are quite a lot of them and it becomes necessary to return to clarify the details. It is also necessary to insert a justification for the use of these particular methods for the purposes of the study.
Response:
Thank you for this helpful recommendation. In response, we have revised the section on research objectives to present them more clearly (Line 54-57, 85-90). Additionally, we have included a visual diagram (Figure 1) to illustrate the study framework and the relationships among the key variables. We have also added justification for the selected analytical methods to align them more explicitly with the study objectives.
Reviewer’s comment 4:
In the discussion of the results, I would like to see practical recommendations that the authors can make based on the results of the study, as well as to outline promising areas for future research.
Response:
Thank you for your insightful suggestion. We have revised the Discussion section to include practical implications based on our findings. Moreover, under each discussion theme, we have added proposed directions for future research (Line 277-282, 292-302, 324-335, 353-366).
Reviewer’s comment 5:
Perhaps, expand the analysis of studies and in the theoretical justification clearly outline the areas of research regarding the study of Covid, factors and consequences on health, which will also slightly expand the list of sources and may be useful for future researchers.
Response:
We appreciate this thoughtful suggestion. In response, we have added a new paragraph in the Introduction to expand on existing research in the field of COVID-19 and its impact on health (Line 324-335, 353-366). This addition also includes several new references (29-31) that we believe will be helpful to future researchers.
Reviewer’s comment 6:
The recommendations presented do not reduce the overall positive impression of the study.
Response:
Thank you again for your kind and supportive feedback.
Reviewer’s comment 7:
The manuscript can be recommended for publication after minor revision.
Response:
We sincerely thank you for your encouraging recommendation and for the valuable time you spent reviewing our work.

Reviewer 5 Report
Comments and Suggestions for Authors
Although COVID has already become a usual virus similar to flue, amny researchers continue to focus on this pandemic and its long-last effects on phesical and mental health. Unfortunately, most of such study don't have reliable data as they started in the middle or even after COVID. Still, some long-lasting effects may take place and as such research in the field remains important. In the present paper there are some comments to be addressed:
- You don't have any rationalle on why would social perceptions of sexual roles affect your results. S it would be reasonable you use term "sex" not "gender" in you manuscript. More over, you categorize your participants as "female" and "male" that also corresponds to sex, no gender.
- You consider sleep problems somehow related to COVID, but your participants are medical students that due to specifics of their education have continuous sleep problems. You need to prove that these sleep isuues are really related to COVID effects.
- It seems that your account for "Use of NRICM10" is totall unjustified. I would exclude it from analysis as it seems having no scienfic ground (not the medicine itself, but the way you analyzed it).
- The biggest limitation of your study that you neder discuss or take in account is that your sample is not just student, they are medical students. This means they have specific routine, academic overload and most importantly, they we close to the desease (COVID) than any other student. That could affect them in totally different way than usual young people, not related to medicine, ehich makes your results ungeneralizable for any other population.
Author Response
Point-by-point Response Letter
Manuscript ID: healthcare-3707168
To the Reviewer:
Thank you very much for taking time from your busy schedule to comment on our manuscript. Our responses to your suggestions and questions are summarized below, with reference to the appropriate pages in the text. Revisions in the text have also been highlighted in RED. We truly appreciate your thoughtful and constructive comments that help make this a better paper. We are very grateful to you for all the comments and insightful suggestions that help enhance both the quality and readability of our paper. Thank you.
Reviewer’s comment 1:
Although COVID has already become a usual virus similar to flue, many researchers continue to focus on this pandemic and its long-last effects on physical and mental health. Unfortunately, most of such study don't have reliable data as they started in the middle or even after COVID. Still, some long-lasting effects may take place and as such research in the field remains important.
Response:
Thank you for your positive evaluation of the study’s relevance. We agree that continued research on long-term COVID-19 effects is necessary, especially as new evidence continues to emerge.
Reviewer’s comment 2:
In the present paper there are some comments to be addressed: You don't have any rationale on why would social perceptions of sexual roles affect your results. S it would be reasonable you use term "sex" not "gender" in you manuscript. Moreover, you categorize your participants as "female" and "male" that also corresponds to sex, no gender.
Response:
Thank you for this correction. We agree with your observation and have revised all instances of “gender” to “sex” in the manuscript to ensure conceptual accuracy and consistency with our data collection categories (male/female) (Line 34, 125, table 1).
Reviewer’s comment 3:
You consider sleep problems somehow related to COVID, but your participants are medical students that due to specifics of their education have continuous sleep problems. You need to prove that these sleep issues are really related to COVID effects.
Response:
Thank you for this important point. We agree that students in medical-related fields are prone to sleep disturbances due to academic pressure. In our study, participants were asked to self-report sleep disturbances within the past month. The proportion reporting no sleep problems was 73.6% among uninfected participants and 67.2% among those who had recovered from COVID-19 more than three months prior. The difference was not statistically significant. This suggests that while approximately one-third of participants experienced sleep disturbances, the observed differences may not be directly attributable to COVID-19 infection. We acknowledge that our self-report measure limits causal interpretation, and we have noted this limitation in the revised Discussion section (Line 371-377).
Reviewer’s comment 4:
It seems that your account for "Use of NRICM10" is total unjustified. I would exclude it from analysis as it seems having no scientist ground (not the medicine itself, but the way you analyzed it).
Response:
Thank you for your observation. In response, we have removed NRICM101 from our analytical results. However, given that the use of TCM is a culturally embedded health behavior in Taiwan, we have retained some basic data and included a brief discussion to highlight the broader health-seeking patterns among the study population (Line 343-348).
Reviewer’s comment 1:
The biggest limitation of your study that you need discuss or take in account is that your sample is not just student, they are medical students. This means they have specific routine, academic overload and most importantly, they we close to the disease (COVID) than any other student. That could affect them in totally different way than usual young people, not related to medicine, which makes your results ungeneralizable for any other population.
Response:
Thank you for this insightful comment. We would like to clarify that although participants were recruited from a medical university, they were not exclusively medical school students. The sample included students from various departments within the university, including public health, pharmacy, nursing, biomedical sciences, and traditional medicine. That said, we agree with your broader point: students from medical-related programs may share characteristics—such as intensive academic schedules and increased clinical exposure—that distinguish them from non-medical student populations. This limits the generalizability of our findings, and we have added this as a limitation in Section 4.4 (Limitations). Furthermore, we have revised Section 3.1 (Demographic Characteristics) to provide detailed information on the distribution of participants across different departments (Line 209-215). Lastly, to prevent confusion, we have revised all references to “medical students” to “medical-related students” throughout the manuscript.

Round 2
Reviewer 2 Report
Comments and Suggestions for Authors
Thank you to the authors for reviewing the manuscript. However, I would like to add a few comments.
Comment 1: I think the abstract doesn't need lines 27-28. "...using independent t-tests, chi-square tests, or Fisher's exact tests."
Comment 2: In my opinion, including the results of statistical calculations, such as p-values and odds ratios (ORs), in the abstract is important.
Comment 3: There is still a lack of information about NRICM101. It is difficult to understand what it is. I think a brief explanation about this drug is needed.
Comment 4: If more than three sources are cited consecutively, they should be listed with a hyphen. For example, in line 82, [11, 12, 13] should be formatted as [11-13].
Comment 5: Fig. 1 Study framework should be under the Figure, not to the side.
Comment 6: Please complete the Data Analysis section. Write down the criteria you used to determine whether the regression model is appropriate. Also, indicate how you assessed the normal distribution of the data.
Author Response
Point-by-point Response Letter Manuscript ID: healthcare-3707168
To the Reviewer:
Thank you very much for taking time from your busy schedule to comment on our manuscript. Our responses to your suggestions and questions are summarized below, with reference to the appropriate pages in the text. Revisions in the text have also been highlighted in RED. We truly appreciate your thoughtful and constructive comments that help make this a better paper. We are very grateful to you for all the comments and insightful suggestions that help enhance both the quality of our paper. Thank you.
Reviewer’s comment 1:
I think the abstract doesn't need lines 27-28. "...using independent t-tests, chi-square tests, or Fisher's exact tests."
Response:
Thank you for your suggestion. We have removed the phrase “using independent t-tests, chi-square tests, or Fisher's exact tests” from the abstract as recommended (Line 26-27).
Reviewer’s comment 2:
In my opinion, including the results of statistical calculations, such as p-values and odds ratios (ORs), in the abstract is important.
Response:
Thank you for your valuable suggestion. We have added the relevant p-values and odds ratios (ORs) to the abstract as follows:
“Logistic regression identified that prior COVID-19 diagnosis was associated with the presence of long COVID symptoms (odds ratio = 1.48, p = .024) in the adjusted model analysis. Meanwhile, higher anxiety levels (odds ratio = 1.09, p < .001) and a BMI ≥ 24 (odds ratio = 4.50, p < .01) emerged as significant risk factors for post-infection syndrome among previously infected students.” (see Lines 32–35; Result, Lines 267–268; and Lines 273–274).
Reviewer’s comment 3:
There is still a lack of information about NRICM101. It is difficult to understand what it is. I think a brief explanation about this drug is needed.
Response:
Thank you for pointing this out. We have added a brief explanation of NRICM101 in the background section of the Introduction to provide context for readers (Lines 81–85).
Reviewer’s comment 4:
If more than three sources are cited consecutively, they should be listed with a hyphen. For example, in line 82, [11, 12, 13] should be formatted as [11-13].
Response:
Thank you for your attention to detail. We have corrected all instances where three or more consecutive citations were listed, using a hyphen format, such as [11–13] (see Line 87; Line 146; and Line 151).
Reviewer’s comment 5:
Fig. 1 Study framework should be under the Figure, not to the side.
Response:
Thank you for the observation. We have repositioned the “Fig. 1 Study framework” title so that it now appears below the figure, as required (see Figure 1).
Reviewer’s comment 6:
Please complete the Data Analysis section. Write down the criteria you used to determine whether the regression model is appropriate. Also, indicate how you assessed the normal distribution of the data.
Response:
Thank you for your suggestion. We have revised the Data Analysis section to explicitly describe how we determined the appropriateness of the regression model and how we assessed the normal distribution of the data. The revised section now reads as follows:
- Normality of continuous variables was assessed using skewness and kurtosis. Group comparisons were conducted using independent t-tests for continuous variables and chi-squared or Fisher's exact tests for categorical variables. The dependent variable—presence of long COVID symptoms—was binary, and predictor variables included both categorical and continuous types; therefore, logistic regression analysis was used to identify associated factors. Model fit was assessed using Nagelkerke’s R², and the interpretation criteria proposed by Hosmer & Lemeshow and MedCalc [21] were applied: R² values between 0.1 and 0.2 indicate modest explanatory power, and values between 0.2 and 0.4 indicate moderate explanatory power. (Lines 200–209)
- We have also included the interpretation of Nagelkerke R² values in the Results section for clarity (Lines 275–279).
- We add a citation in content and reference list for the interpretation of Nagelkerke R² values (L207 and reference 21).

Reviewer 3 Report
Comments and Suggestions for Authors
Thank you for revising and adding content based on the reviewer comments within such a short period of time. I have confirmed that the comments have been appropriately addressed and reflected in the manuscript.
Best regards,
Author Response
Point-by-point Response Letter
Manuscript ID: healthcare-3707168
Reviewer’s comment 1:
Thank you for revising and adding content based on the reviewer comments within such a short period of time. I have confirmed that the comments have been appropriately addressed and reflected in the manuscript.
Response:
Thank you very much for taking the time to review our manuscript. We are grateful for your encouraging feedback and thoughtful suggestions. Your positive evaluation and constructive recommendations have helped us refine the manuscript and strengthen the clarity of our study. We sincerely thank you for your encouraging recommendation and for the valuable time you spent reviewing our work.

Reviewer 5 Report
Comments and Suggestions for Authors
Thank you for your clarifications and for this research.
Author Response
Point-by-point Response Letter
Manuscript ID: healthcare-3707168
Reviewer’s comment 1:
Thank you for your clarifications and for this research.
Response:
Thank you very much for taking time from your busy schedule to comment on our manuscript. We truly appreciate your thoughtful and constructive comments that help make this a better paper. We are very grateful to you for all the comments and insightful suggestions that help enhance both the quality and readability of our paper. Thank you.
